# Analysis of Nuclear Architecture and Chromatin Organization in SXT Data

**Ayse S. Erozan**[1,2,3] iD                    AYSE.EROZAN@KIT.EDU
**Fariha M. Annesha**[1,3]                    FARIHA.ANNESHA@KIT.EDU
**Vincent Heuveline**[2]                    VINCENT.HEUVELINE@UNI-HEIDELBERG.DE
**Venera Weinhardt**[1]                    VENERA.WEINHARDT@KIT.EDU
[1] *Institute of Microstructure Technology, Karlsruhe Institute of Technology, Karlsruhe, Germany*
[2] *Interdisciplinary Center for Scientific Computing, Heidelberg University, Heidelberg, Germany*
[2] *Centre for Organismal Studies, Heidelberg Univeristy, Heidelber, Germany*

## Abstract

Accurate characterization of the nuclear architecture is critical in cell biology, particularly in cancer research. Abnormal nuclear morphology and altered chromatin organization are widely recognized as key indicators of malignancy. These structural changes are present as variations in intensity and texture in the imaging data. In this context, soft X-ray tomography (SXT) enables high-resolution, three-dimensional imaging of whole cells, providing a powerful approach for the quantitative analysis of nuclear structures. Building on our previous work, we extend the developed 3D nucleus segmentation to the analysis of the DNA packing. This integrated approach facilitates an improved analysis of intranuclear organization and offers new opportunities to study nuclear abnormalities in cancer.

**Keywords:** Nucleus, automated 3D segmentation, chromatin, single-cell, SXT.

## 1. Introduction

Alterations in nuclear structure are a well-established hallmark of cancer (Gridina and Fishman, 2022; Zink et al., 2004) and infectious disease(Loconte et al., 2021; Chen et al., 2022), and are widely used in clinical diagnosis. Tumor cells often exhibit increased nuclear size, irregular nuclear contours, and changes in chromatin texture (Uhler and Shivashankar, 2018). These morphological changes are sufficiently distinctive to allow to distinguish between benign and malignant cells and to classify tumor types and stages (Zink et al., 2004).

Inside the nucleus, chromosomes are arranged in a highly organized three-dimensional structure rather than as linear entities. This organization includes chromosome territories and spatially distributed chromatin regions, which can be broadly categorized into euchromatin and heterochromatin (Gridina and Fishman, 2022). Euchromatin is generally less condensed and associated with active genomic regions, whereas heterochromatin is more compact and typically found in transcriptionally inactive regions (Morrison and Thakur, 2021). The spatial arrangement of these chromatin states is closely related to nuclear function (Pongubala and Murre, 2021). In cancer cells, this organization is frequently disrupted, leading to visible changes in chromatin distribution and nuclear morphology. Such alterations can be observed in imaging data as differences in chromatin texture and spatial organization (Zink et al., 2004; Amodeo et al., 2025; Easwaran and Baylin, 2011).

Given the importance of nuclear morphology and chromatin organization in cancer characterization, accurate segmentation of the nucleus and its internal structures is essential for

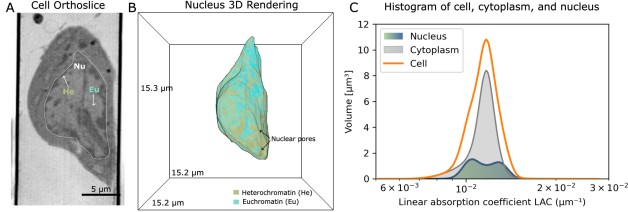

Figure 1: Detailed nuclear architecture visualized by soft X-ray tomography imaging (A) A 2D orthoslice from SXT data(B) 3D reconstruction of the nucleus, Adapted figure from Erozan et al. (Erozan et al., 2025) (C) Histogram of the cell (orange line), the cytoplasm (gray), and (blue-to-green gradient)

analysis. Reliable segmentation enables the extraction of morphological features such as nuclear size, shape, and chromatin distribution, which are critical for downstream analysis. In this work, we extend our previous approach for automatic nucleus segmentation in SXT data by incorporating chromatin segmentation, enabling a more detailed qualitative assessment of nuclear organization.

## 2. Methods

**Dataset** We conducted our experiments on the SXT dataset, which comprises two distinct cell types: human Jurkat T-cells and mouse microglia (BV-2) cells (see Fig. 1-A). Based on the experimental results reported in our previous work, 11 T-cell and 5 BV-2 cell tomograms were used for training, while 10 T-cell and 10 BV-2 cell tomograms were reserved for testing.

**Nucleus Segmentation Models** For nucleus segmentation, we explored multiple strategies, including active learning, context-awareness, different architectures, and an ablation study. Ultimately, the best performance was achieved with a 3D U-Net (Çiçek et al., 2016) with five levels of depth, initialized with 64 filters. Detailed descriptions of these approaches can be found in our previous work (Erozan et al., 2025).

**Chromatin Segmentation** Following our automated nucleus segmentation, chromatin structures were separated using K-means algorithm. Due to the complex and heterogeneous nature of chromatin as shown in the Figure 1-B, accurate manual delineation would be highly challenging. However, the distribution of tomogram intensities within the nucleus reveals distinct regions corresponding to euchromatin and heterochromatin. These distributions provide a basis for separating chromatin regions in an unsupervised manner. Therefore, we formulated this task as a clustering problem, where K-means with $K = 3$ yielded stable, and consistent separation of chromatin regions. This distinction is illustrated in Fig. 1-C.

## 3. Results

Despite the limited availability of training data, the proposed approach achieved a Dice similarity coefficient of 92.47% for nucleus segmentation, demonstrating robust performance under data-constrained conditions. This result was obtained on a test set of 20 tomograms

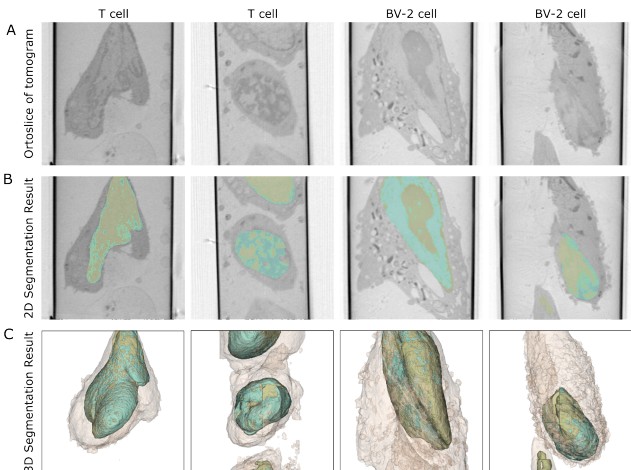

Figure 2: Soft X-ray tomography data along with the corresponding 2D and 3D segmentation results.

across two distinct cell types. These results indicate the method's ability to generalize across varying cellular structures.

For chromatin segmentation, K-Means clustering produced consistent and structurally meaningful partitioning of chromatin regions. As illustrated in Fig. 2, the method effectively captures the global separation of chromatin across multiple cells. This highlights consistent patterns despite structural variability. Notably, differences in chromatin organization were observed between cell types. In T cells, chromatin intensity distributions exhibited a bimodal profile, with a relatively balanced representation of chromatin regions. In contrast, BV-2 cells showed a less pronounced bimodal distribution, with a comparatively lower abundance of heterochromatin. Despite these differences, K-Means clustering consistently produced meaningful separation of chromatin regions in both cases, demonstrating its robustness across varying chromatin distributions.

While some limitations remain, particularly in regions with ambiguous intensity boundaries, the approach preserves the primary structural organization of chromatin. Overall, the segmentation reveals distinguishable patterns that support qualitative analysis and visual interpretation of chromatin distribution.

## 4. Conclusion

In this study, we addressed the analysis of nuclear architecture in SXT data by extending nucleus segmentation with chromatin segmentation. Building our previous framework, we demonstrated that meaningful chromatin patterns can be reliably extracted using unsupervised methods, even in the absence of labeled data.

While the current findings are primarily qualitative, the proposed approach lays the groundwork for future advances in chromatin segmentation and the development of quantitative analysis frameworks.

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
