# OpenReview forum: "Analysis of Nuclear Architecture and Chromatin Organization in SXT Data"
_MIDL.io/2026/Short_Papers — MIDL 2026 - Short Papers Poster_

### Official Review · Reviewer_12sq · 2026-04-23
**important work with qualitative evaluation**

**Rating:** 3
**Confidence:** 4

**Review:**

Accurate characterization of nuclear architecture and chromatin organization is relevant to cancer biology and soft X-ray tomography is a compelling modality for this purpose. Another positive aspect is the simplicity and accessibility of the proposed pipeline which is easy to understand and avoids the need for difficult manual chromatin annotations. The manuscript is clearly written. The main weakness is that the chromatin segmentation component lacks quantitative validation. The claims of meaningful and consistent partitioning are supported primarily through visual examples rather than objective evaluation. Relatedly, the methodological justification for using K-means and especially for choosing K=3 is limited. The paper would also be stronger with either comparisons to alternative unsupervised approaches or at least some expert-based sanity check to support the qualitative findings.

**Summary:**

This paper presents an extension of the authors’ prior automated 3D nucleus segmentation work for X-ray tomography data to also analyze chromatin organization within the nucleus. After segmenting nuclei with a previously developed 3D U-Net, the authors apply unsupervised K-means clustering to partition intranuclear intensity patterns into chromatin regions, motivated by the difficulty of obtaining reliable manual annotations. The paper reports strong nucleus segmentation performance from prior work and shows qualitative chromatin segmentation results across two cell types.

**Strengths:**

- Accurate characterization of nuclear architecture is important for cancer research. Abnormal nuclear organization is associated with malignancy.
- The paper builds upon a previous automated nuclei segmentation by the authors to achieve automated chromatin segmentation using unsupervised k-means algorithm. The approach avoids requiring manual chromatin annotations which is a strength.
- The method is simple and easy to understand: automated nucleus segmentation followed by unsupervised K-means clustering for chromatin partitioning. This makes the pipeline accessible and potentially reproducible.
- The paper is well written and easy to follow.

**Weaknesses:**

- The paper lacks a quantitative validation of the automatically produced results. The paper argues that the outputs are “meaningful” and “consistent,” but this is supported only qualitatively.
- The choice of K-means, and especially of K=3, is not sufficiently justified beyond empirical stability. The paper would be stronger with a clearer biological or methodological explanation of what the three clusters represent and why this is the right granularity.
- While the general motivation for cancer research is discussed, the paper could benefit from a more specific discussion of the types of downstream tasks the proposed method can allow. Because this is an incremental extension of prior nucleus segmentation work, the paper should better clarify what new biological or analytical insight is enabled specifically by the added chromatin segmentation stage.

**Justification Of Rating:**

Overall, the paper is clearly motivated and easy to follow, but the chromatin analysis remains largely qualitative and would benefit from stronger validation and clearer discussion of downstream utility.

---

### Decision · Program_Chairs · 2026-05-08

Accept (Poster)